# Modeling hepatitis C virus kinetics during liver transplantation reveals the role of the liver in virus clearance

**Louis Shekhtman**[1,2†], **Miquel Navasa**[3†], **Natasha Sansone**[1,4†], **Gonzalo Crespo**[3], **Gitanjali Subramanya**[1], **Tje Lin Chung**[1,5], **E Fabian Cardozo-Ojeda**[1,6], **Sofía Pérez-del-Pulgar**[3], **Alan S Perelson**[7], **Scott J Cotler**[1], **Xavier Forns**[3], **Susan L Uprichard**[1,8]*, **Harel Dahari**[1]*

[1]The Program for Experimental & Theoretical Modeling, Division of Hepatology, Department of Medicine, Stritch School of Medicine, Loyola University Medical Center, Maywood, IL, United States; [2]Network Science Institute, Northeastern University, Boston, MA, United States; [3]Liver Unit, Hospital Clínic, IDIBAPS and CIBEREHD, University of Barcelona, Barcelona, Spain; [4]Department of Microbiology & Immunology, University of Illinois Chicago, Chicago, IL, United States; [5]Institute for Biostatistics and Mathematical Modeling, Department of Medicine, Goethe Universität Frankfurt, Frankfurt, Germany; [6]Vaccine and Infectious Disease Division, Fred Hutchinson Cancer Research Center, Seattle, WA, United States; [7]Theoretical Biology and Biophysics, Los Alamos National Laboratory, Los Alamos, NM, United States; [8]The Infectious Disease and Immunology Research Institute, Stritch School of Medicine, Loyola University Medical Center, Maywood, IL, United States

**\*For correspondence:**
suprichard@luc.edu (SLU);
hdahari@luc.edu (HD)

[†]These authors contributed equally to this work

**Abstract** While the liver, specifically hepatocytes, are widely accepted as the main source of hepatitis C virus (HCV) production, the role of the liver/hepatocytes in clearance of circulating HCV remains unknown. Frequent HCV kinetic data were recorded and mathematically modeled from five liver transplant patients throughout the anhepatic (absence of liver) phase and for 4 hr post-reperfusion. During the anhepatic phase, HCV remained at pre-anhepatic levels ($n$ = 3) or declined ($n$ = 2) with $t_{1/2}$~1 hr. Immediately post-reperfusion, virus declined in a biphasic manner in four patients consisting of a rapid decline ($t_{1/2}$ = 5 min) followed by a slower decline ($t_{1/2}$ = 67 min). Consistent with the majority of patients in the anhepatic phase, when we monitored HCV clearance at 37 °C from culture medium in the absence/presence of chronically infected hepatoma cells that were inhibited from secreting HCV, the HCV $t_{1/2}$ in cell culture was longer in the absence of chronically HCV-infected cells. The results suggest that the liver plays a major role in the clearance of circulating HCV and that hepatocytes may be involved.

## Editor's evaluation

In their work, the authors combine clinical data and mathematical modelling to shed light on the role of hepatocytes in HCV clearance. This manuscript will be of interest to clinicians in organ transplantation centers and to translational hepatitis virus researchers given that it provides a rare and carefully collected dataset of hepatitis C virus blood titers during and after liver transplantation. The manuscript is also of potential interest to modelers interested in HCV infection and more broadly infectious disease specialists.

**Figure 1.** Virus kinetics in the serum of five hepatitis C virus (HCV)-infected patient before, during, and after liver transplantation (LT). (a) Surgery was initiated at $t = 0$. The anhepatic phase is the first lighter shading and the darker shading immediately after it represents the first 4 hr post-reperfusion (RP). In all patients, virus levels decreased after graft reperfusion. A nadir was reached after a median time of 15 hr (range 9–87 hr) post-reperfusion. HCV RNA resurged within 7 days after reperfusion, with three of five cases reaching greater or comparable viral levels to those observed at baseline. For the remaining two cases, the corresponding viral loads were 0.87 and 0.43 $\log_{10}$ IU/mL lower than those just before surgery. HCV RNA levels in Case 5 remained in a lower plateau from 11 hr post-surgery until the end of follow-up period (slope <0.001, p = 0.76), with a corresponding viral load of 2.45 ± 0.002 $\log_{10}$ IU/mL. (b) A zoomed-in version of (a) focusing on the time of LT and 4 hr post-RP.

## Introduction

The liver, specifically hepatocytes, is widely accepted as the main source of hepatitis C virus (HCV) production but the role of the liver/hepatocytes in the clearance of circulating HCV remains largely unknown. To determine the function of the liver/hepatocytes in clearing cell-free virus from the circulation in patients requires being able to monitor and compare viral kinetics during the presence and absence of the liver/hepatocytes. This can be achieved readily in vitro but only transiently in vivo during liver transplantation where viral levels can be measured in the blood during the anhepatic phase when the native liver has been removed compared to after reperfusion of the new liver.

Previous kinetic studies (*Dahari et al., 2005*; *Fukumoto et al., 1996*; *Garcia-Retortillo et al., 2002*; *Hughes et al., 2017*; *Powers et al., 2006*) have suggested that HCV was cleared at the same rate during the anhepatic phase and early after reperfusion. However, these estimates were based on limited data ( one to four samples during the anhepatic phase), obtained at the beginning and end of the anhepatic phase and several hours after reperfusion. Interestingly, *Garcia-Retortillo et al., 2002*, reported a patient with a prolonged anhepatic phase of 20 hr in whom the HCV half-live ($t_{1/2}$) during the anhepatic phase and after reperfusion was estimated to be 10.3 and 3.8 hr, respectively, suggesting that viral clearance occurs relatively slowly during the anhepatic phase and increases after

**Table 1.** Characteristics of liver transplant (LT) recipients/donors and fluid balances during transplantation.
BMI is the body mass index defined as a ratio of weight in kg to the square of height in meters. AH, anhepatic phase; RP, 4 hr after graft reperfusion.

| Case | Gender | Weight (kg) | BMI (kg/m²) | Age at LT (year) | Donor age (year) | Donor gender | HCV genotype | AH fluid intake (mL) | RP fluid intake (mL) |
|------|--------|-------------|-------------|------------------|------------------|--------------|--------------|----------------------|----------------------|
| 1 | M | 88 | 30 | 68 | 72 | M | 1b | 0 | 0 |
| 2 | F | 70 | 31 | 66 | 33 | M | 1b | 500 | 1000 |
| 3 | M | 87 | 34 | 56 | 69 | M | 3 | 250 | 250 |
| 4 | F | 43 | 19 | 60 | 73 | F | 1b | 750 | 3300 |
| 5 | M | 90 | 29 | 49 | 29 | M | 1b | 951 | 3350 |
| Median (range) | | 87 (43–90) | 30 (19–34) | 60 (49–68) | 69 (29–73) | | | 500 (0–951) | 1000 (0–3350) |

reperfusion. *Garcia-Retortillo et al., 2002*, also hypothesized that the rapid clearance of HCV from the serum post-reperfusion was at least in part due to entrance of HCV into the hepatocytes or the liver reticuloendothelial system. However, a detailed viral kinetic study during the anhepatic phase and immediately after reperfusion has not been performed.

Here, we measured viral levels in five liver transplant patients over the course of liver transplantation (*Figure 1a*). We focus on very frequent sampling during the anhepatic phase (i.e., every 5–15 min) and during the first 4 hr of the reperfusion period (every 5–10 min) (*Figure 1b*) along with recording fluid balances during the surgical procedure (*Table 1*). The viral clearance rate during these two distinct phases was estimated using a mathematical modeling framework (*Figure 2*) accounting for viral kinetics and taking into account fluid balance including infusion of red blood cells, plasma, saline, and albumin, as well as blood loss. HCV serum levels were observed to remain at steady state during the AH phase in three of the five patients suggesting that the liver plays a major role in clearing HCV from the circulation. Interestingly, in parallel in vitro experiments, when we monitored the clearance rate of HCV from culture medium at 37 °C in the absence and presence of chronically infected Huh7 human hepatoma cells, HCV clearance was faster in the presence of the hepatoma cells suggesting that hepatocytes may play a role in the observed clearance of circulating HCV.

## Results

### Viral kinetics before and during the anhepatic phase

Prior to the anhepatic phase, the viral load is at steady state. Since there was minimal variation prior to the anhepatic phase, we defined the final measured value of viral load prior to the anhepatic phase as the initial viral load for the model. The median viral load measured just before the anhepatic phase was 6.0 (range 3.7–6.4) $\log_{10}$ IU/mL (*Figure 3*; *Table 2*). We carried out a linear fit to the data during the anhepatic phase, which lasted 1.25–1.90 hr. We found that virus levels were flat in three cases (Case 1, Case 2, and Case 3) while in two patients (Case 4 and Case 5) the viral decline slope was 0.32 and 0.16 $\log_{10}$ IU/mL per hour, respectively (*Table 2*).

### Viral kinetics after graft reperfusion

During the first 4 hr after reperfusion, HCV measurements were taken at intermittent times (*Figure 3*) and volume input was recorded. Fluid input during post-reperfusion for Cases 1–5 were 0, 1000, 250, 3300, and 3350 mL, respectively (*Table 1*). In Cases 1–4, HCV RNA declined in a biphasic manner, while in Case 5, HCV RNA remained at the same levels as at the end of the anhepatic phase. For Cases 1–4, the biphasic decline consisted of an initial, sharp decrease within 11–22 min, followed by a longer but slower decrease (*Table 3*). Using linear regression, the slope of the initial decrease was 2.7, 1.5, 2.1, and 2.0 $\log_{10}$ IU/mL per hour for Cases 1–4, respectively, that lasted for 1.2–2.8 min. The slower second phase slope for Cases 1–4 was 0.2, 0.3, 0.2, and 0.4 $\log_{10}$ IU/mL per hour, respectively (*Table 3*). The viral plateau in Case 5 is distinct from the other cases, although it is worth noting that this patient's viral load continued to be flat for ~6 days post-reperfusion (*Figure 1*).

### Modeling HCV kinetics

Model fits are shown in *Figure 3* and viral clearance ($c_{AH}$) estimates are shown in *Table 4*. The estimated virus clearance rates during the anhepatic phase in Cases 1–3 were ~0 (i.e., very long $t_{1/2}$ that implies no clearance) but were significantly different from 0 in Cases 4 and 5 corresponding to an HCV $t_{1/2} = \text{Ln}(2)/c_{AH}$ of 1–2 hr. Similar results were obtained under the assumption of 5 L of extracellular fluid per 70 kg (*Supplementary file 1*, Table S1).

When fitting the model to data from the reperfusion phase, we found that there was significant correlation between the parameters $c_0$ and $c_{RP}$, representing the initial faster clearance rate, and the longer term slower clearance rate, respectively. The high correlation between the parameters suggests that these two parameters were not independently identifiable. This led us to fit the model in two steps. first, we set all of the parameters $c_0$, $c_{RP}$, and $\kappa$, which governs the speed of the transition from $c_0$ to $c_{RP}$, as free. Second, we fixed $c_0$ to its best-fit value and then fit the model again estimating $\kappa$, $c_{RP}$. For Cases 1–4, the initial viral clearance, $c_0$, gave $t_{1/2}$ on the order of 5 [0.4–16] min (*Table 5*). The median time constant was $\kappa = 399$/day meaning that the time it took for the clearance rate to shift from its initial value of $c_0$ to its final value of $c_{RP}$ was around 4 min. The second, slower clearance

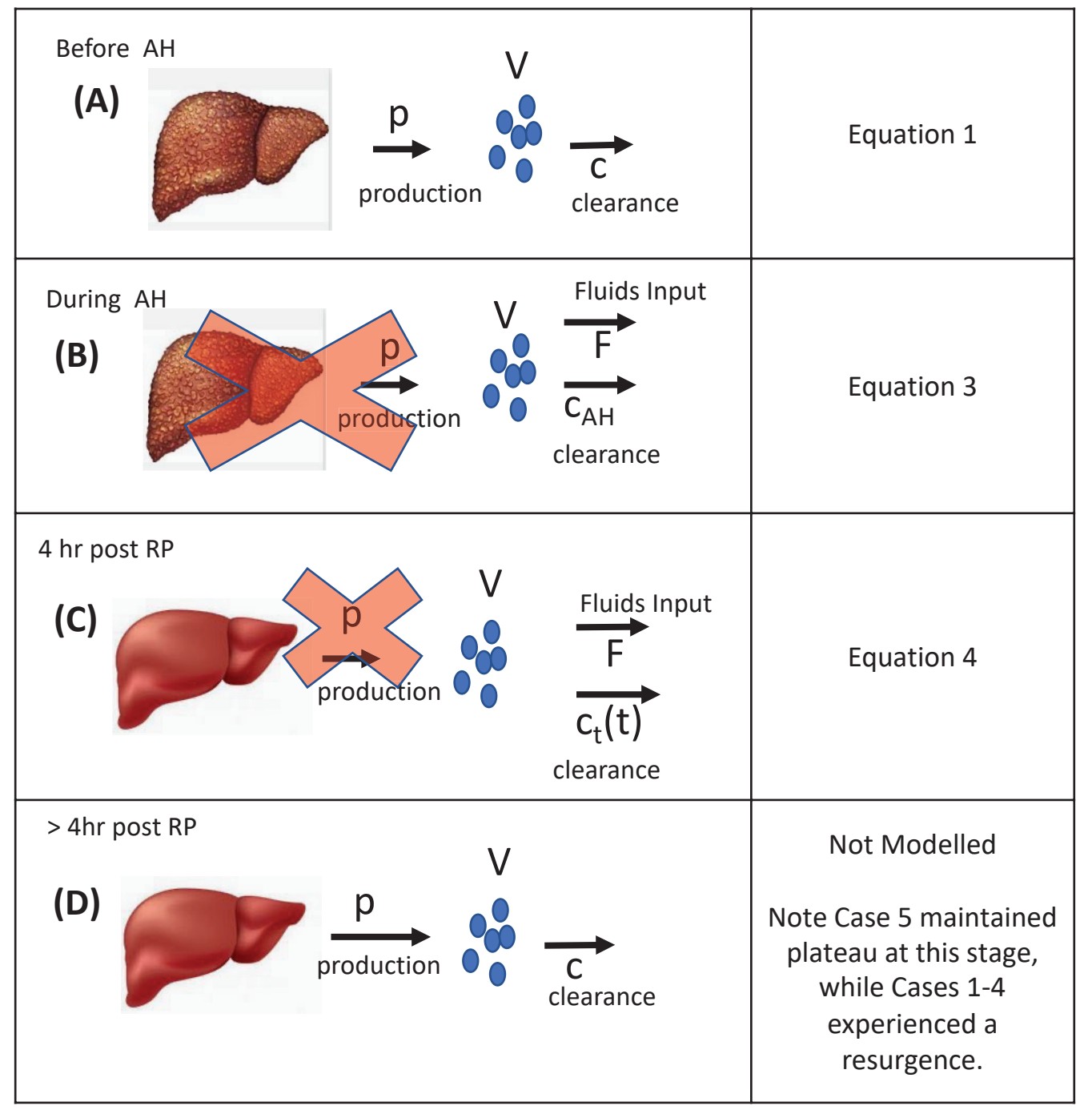

**Figure 2.** Visual description of the stages of the transplant procedure and modeling assumptions. (**A**) Prior to the anhepatic (AH) phase viral load is at a steady state in which virus production is balanced by viral clearance (**Equation 1**). (**B**) During the AH phase, the liver is not present and therefore there is no viral production (**Equation 2**). However, virus may appear to be cleared by the input of fluids that dilute the virus in circulation (see *F* in **Equation 3**) and through processes of *physiological* clearance. (**C**) Up to 4 hr post-reperfusion there is no viral production as the cells in the new liver if they have been infected have not yet started releasing new virions. Clearance still occurs both via fluid balances but is predicted via a time-dependent function (see *c(t)* in **Equation 4**) that may represent a rapid clearance phase (possibly due to viral binding/entry into hepatocytes) immediately after graft reperfusion and a second/slower clearance phase (*physiological*). (**D**) Longer after reperfusion (over 4 hr ), the new liver begins producing new virions (as evident in **Figure 1a** in Cases 1–4).

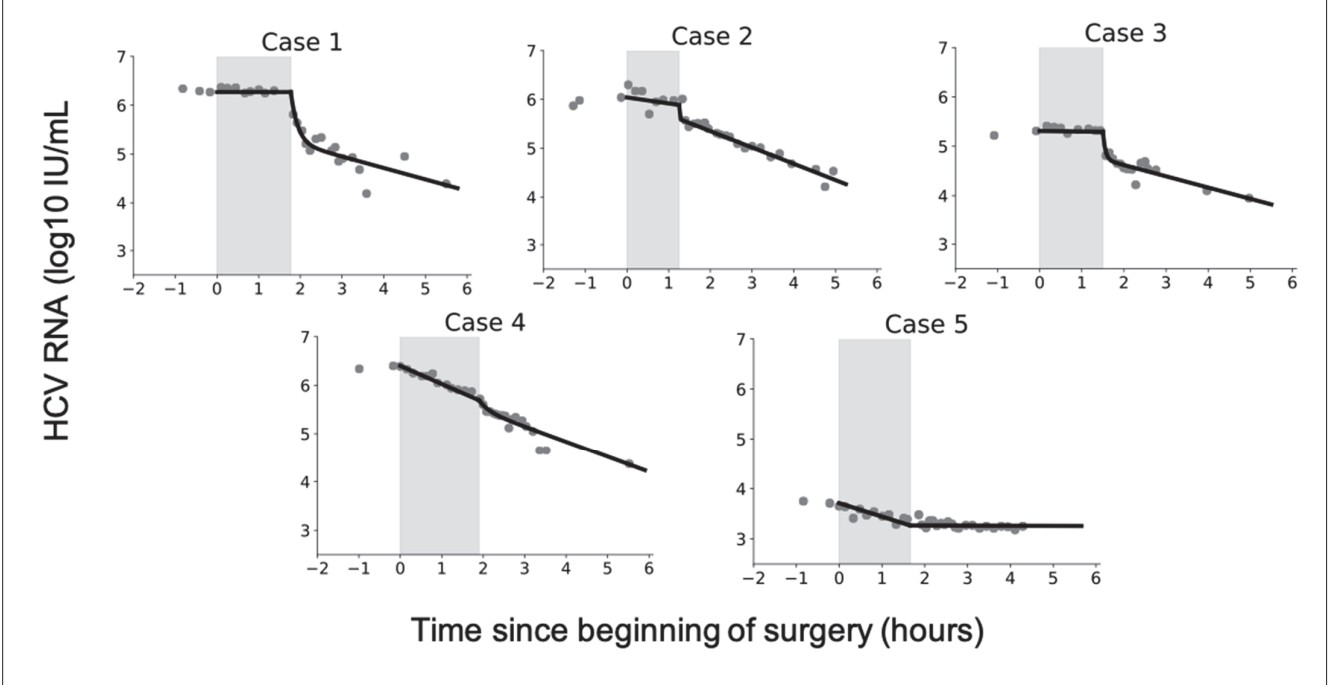

**Figure 3.** Hepatitis C virus (HCV) RNA kinetics during liver transplantation. Serum HCV RNA kinetics in five cases before transplant, during the anhepatic phase (gray rectangles), and during the first 4 hr after liver graft reperfusion. HCV RNA measurements are shown by gray circles and best-fit model predictions by solid black lines. HCV levels are graphed relative to the time of liver removal ($t = 0$) on the x-axis.

The online version of this article includes the following figure supplement(s) for figure 3:

**Source data 1.** Hepatitis C virus (HCV) RNA measurements during liver transplantation.

rate, $c_{RP}$, corresponded to a $t_{1/2}$ of order 67 [54–79] min (**Table 5** and **Figure 3**). In Case 5, **Equation. (3)** was used because an extremely slow (or plateaued) viral load (slope of 0.06 $\log_{10}$ IU/mL/hr) was observed suggesting no production and clearance during the first 6 hr post-reperfusion (**Figure 3**, **Table 5**) or the subsequent 6 days (**Figure 1**). Similar results were obtained under the assumption of 5 L of extracellular fluid per 70 kg (**Supplementary file 2**, Table S2).

## HCV clearance from culture medium in vitro

To further investigate the role of hepatocytes in HCV clearance, we turned to the more controlled in vitro HCV infection experimental system (**Zhong et al., 2005**) as previously described in detail (**Yu and Uprichard, 2010**). To determine the effect of Huh7 hepatoma cells on HCV clearance in vitro,

**Table 2.** Hepatitis C virus (HCV) kinetics during anhepatic phase.

The viral load prior to the anhepatic phase (AH) is the final measurement just before the anhepatic phase. The decrease was determined by linear regression.

| Phase | Case | Viral load prior to AH (log IU/mL) | Decrease slope (log/hr) | p-Value |
|---|---|---|---|---|
| AH | 1 | 6.3 | 0.07 | 0.073 |
| | 2 | 6.0 | 0.30 | 0.17 |
| | 3 | 5.3 | 0.06 | 0.16 |
| | 4 | 6.4 | 0.32 | 1.3e-7 |
| | 5 | 3.7 | 0.16 | 0.006 |
| Median (range) | | 6.0 (3.7–6.4) | 0.16 (0.05–0.32) | |

**Table 3.** Hepatitis C virus (HCV) kinetics during 4 hr post-reperfusion.

| Phase | Case | Viral load prior to RP* (log IU/mL) | Initial decrease slope (log/hr)† and (duration, min) | Second Decrease slope (log/hr) ‡ |
|---|---|---|---|---|
| RP | 1 | 6.3 | 2.7 (16) | 0.25 |
| | 2 | 6.0 | 1.5 (22) | 0.35 |
| | 3 | 5.3 | 2.1 (14) | 0.22 |
| | 4 | 5.9 | 2.0 (11) | 0.37 |
| | 5 § | 3.4 | – | 0.06 |
| Median (range) | | 5.9 (3.4–6.3) | 2.0 (1.5–2.8) | 0.25 (0.06–0.37) |

*The viral load prior to reperfusion is the final measurement during the anhepatic phase just before reperfusion.
†Slopes were calculated using the measured viral load (VL) of the last point of anhepatic phase and the first 4four points of reperfusion phase.
‡VL after the first 4four points of reperfusion until 4 hr post -reperfusion were used. All slopes (determined by linear regression) were significantly different than 0 (Pp ≤ 0.01).
§For Case five5 during graft reperfusion (reperfusion) only one slope was estimated from the last point of anhepatic phase until 4 hr post -reperfusion (RP);. For Cases 1–4 during reperfusion, an initial rapid decrease was estimated, followed by a slower/second decrease.

we measured the half-life of HCV virions at 37 °C in cell culture medium in the absence and presence of chronically infected Huh7 cells by monitoring the decrease of HCV RNA over time (**Figure 4**). To measure HCV half-life in cell culture medium in the presence of chronically HCV-infected cells, we established a steady-state chronic infection in Huh7 cells, and then inhibited the secretion of HCV from the cells using 1 nM of the NS5a inhibitor daclatasvir to block HCV replication and secretion (**Meanwell and Belema, 2019**) or 200 µM naringenin to block HCV secretion (**Goldwasser et al., 2011**). In the absence of cells, the HCV half-life observed was 112 hr (**Figure 4a**). In the presence of chronically infected cells inhibited from de novo virus secretion, the HCV half-life observed was 13 hr (**Figure 4b**), which is 8.5 times faster than the clearance observed in the absence of cells (**Figure 4a**).

## Discussion

The current cross-disciplinary study employs in vivo clinical methods and mathematical modeling to assess virus levels during the anhepatic phase and immediately after graft reperfusion, providing

**Table 4.** Best-fit parameter estimates determined by fitting **Equation. (3)** to data obtained during the anhepatic (AH) phase (assuming extracellular fluid of 15 L per 70 kg), where we assumed that fluid intake and outtake are equal (see Materials and methods).
VP, Due to viral plateau (not significantly different from slope 0 as indicated in **Table 2**) hepatitis C virus (HCV) half-life ($t_{1/2}$) was not undefined. Note that in the model all parameter values are converted to mL, but for simplicity some are written here in terms of L. $V_{0AH}$ is the final measured value prior to the AH phase.

| Case | Phase duration (days) | Num. data points | i (fluid intake/ phase duration) (L/day) | F(0) (weight*15 L/70 kg) (L) | $V_{0AH}$ (log IU/mL) | HCV $t_{1/2}$ (ln(2)/$c_{AH}$) (min) (95% CI) |
|---|---|---|---|---|---|---|
| 1 | 0.074 | 8 | 0 | 18.9 | 6.3 | VP |
| 2 | 0.052 | 7 | 9.6 | 15.0 | 6.0 | VP |
| 3 | 0.062 | 8 | 4.0 | 18.6 | 5.3 | VP |
| 4 | 0.079 | 12 | 9.5 | 9.2 | 6.4 | 57 (54–60) |
| 5 | 0.069 | 11 | 13.7 | 19.3 | 3.7 | 79 (67–95) |
| Median (range) | | | 0.067 (0.052–0.079) | 8 (7–12) | 9.5 (0–13.7) | 18.6 (9.2–19.3) | 6.0 (3.7–6.4) |

**Table 5.** Best-fit parameter estimates determined by fitting *Equation. (4)* to data obtained during the 4 hr after graft reperfusion (assuming extracellular fluid of 15 L per 70 kg), where we assumed that fluid intake and outtake are equal (see Materials and methods).

| Case | Phase duration (days) | Num. data points | i (fluid intake/ outtake duration) (L/day) | F(0) (weight*15 L/70 kg) (L) | $V_{ORP}$ (log IU/mL)[†] | Init. HCV $(\ln(2)/c_0)t_{1/2}$[*] (min) | $\kappa$ (1/days) (95% CI) | HCV $t_{1/2}$ $(\ln(2)/c_{RP})$ (min) (95% CI) |
|---|---|---|---|---|---|---|---|---|
| 1 | 0.17 | 15 | 0 | 18.9 | 6.3 | 2.9 | 143 [120–166] | 77 [53–140] |
| 2 | 0.17 | 22 | 6.0 | 15.0 | 5.9 | 0.4 | 3620 [3080–4,150] | 54 [50–59] |
| 3 | 0.17 | 15 | 1.5 | 18.6 | 5.3 | 1.9 | 399 [348, 450] | 79 [63–105] |
| 4 | 0.17 | 18 | 19.8 | 9.2 | 5.7 | 15.8 | 100[‡] | 59 [52–68] |
| 5 | 0.17 | 21 | 20.1 | 19.3 | 3.3 | N/A | N/A | §  |
| **Median (range)** | **0.17** (0.17–0.17) | **18** (15–22) | **6.0** (0–20.1) | **18.6** (9.2–19.3) | **5.7** (3.3–6.3) | **2.4** (0.4–15.8) | **399** (143–3620) | **68** (54–79) |

[*]Since $c_0$ and $c_{RP}$ were highly correlated and not independently identifiable (correlation matrix) and population modeling (using Monolix) indicated that $c_0$ was not identifiable (not shown), the initial virus clearance rate (i.e., $c_0$ in *Equation. 4*) was fixed to its best-fit value (first estimated with $c_0$, $c_{RP}$, and $\kappa$ as free parameters) and then the errors on the remaining parameters ($c_{RP}$ and $\kappa$) were computed.

[†]For 4 hr after reperfusion, the $V_{ORP}$ listed is the final value of the fit for anhepatic phase. This value was then used in the model for calibration with HCV data 4 hr after reperfusion.

[‡]Parameter was set to this fixed value due to high uncertainty and was omitted from the calculation of median and range.

§*Equation. 3* was used to estimate the HCV $t_{1/2}$. Since the best estimate of clearance was $c_{RP}$~0, half-life is undefined.

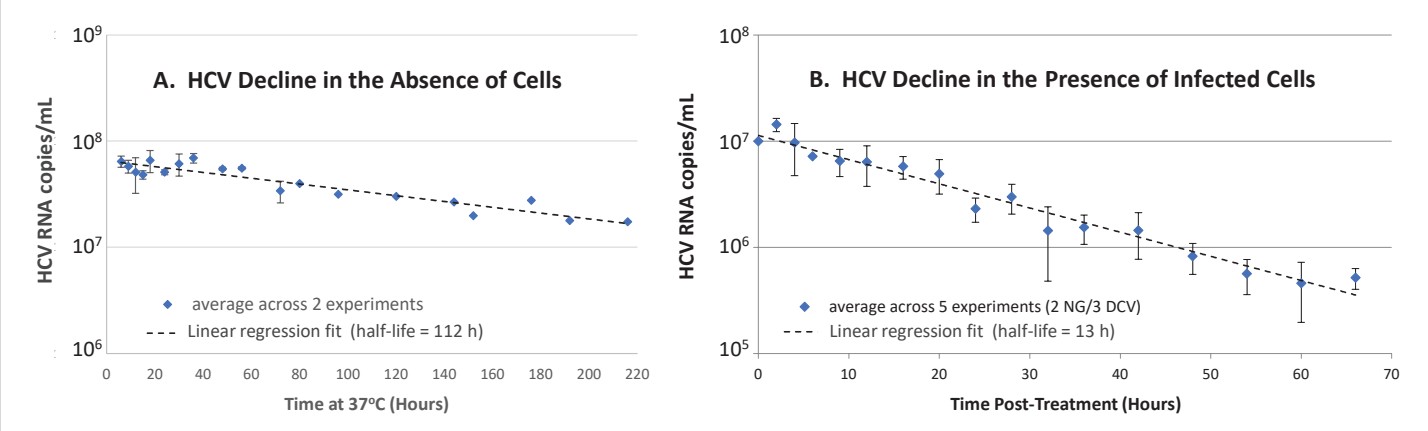

**Figure 4.** In vitro hepatitis C virus (HCV) RNA clearance kinetics. HCV RNA clearance from culture medium at 37°C was monitored by frequent sampling at the indicated time points. (**A**) Decline in HCV RNA in the absence of cells averaged across two experiments, each performed with four replicate wells per time point. (**B**) Decline in HCV RNA in the presence of chronically infected cells in which de novo HCV secretion is inhibited. Chronically HCV-infected non-growing Huh7 cells were mock-treated or treated with 200 μM of the HCV secretion inhibitor naringenin (NG) in two experiments performed with duplicate wells or 1 nM of the HCV replication and secretion inhibitor daclatasvir (DCV) in three experiments, two performed with duplicate wells, one performed in singlet. The average copies/mL ± standard deviation across these five experiments are graphed (blue diamonds). For all experiments, RNA was extracted, and HCV copies were quantified by RT-qPCR in technical duplicate. For the infection experiment, HCV copies/ mL were normalized to mock-treated samples at each time point. Graphed is the average copies/mL ± standard deviation across experiments at the indicated times (blue diamonds). Linear regression (dashed lines) was performed to calculate the virus half-life ($t\frac{1}{2}$).

The online version of this article includes the following figure supplement(s) for figure 4:

**Source data 1.** Hepatitis C virus (HCV) RNA measurements in vitro.

evidence that the liver plays a major role in HCV clearance from the circulation. While others have suggested that the liver sinusoidal endothelial may be responsible for clearance of other viruses from circulation (*Ganesan et al., 2011*), our in vitro data also implicates hepatocytes as possibly playing a role, opening up a new avenue for investigation.

To enhance our ability to derive accurate viral kinetic data over previous studies, very frequent sampling for viral load monitoring during both the anhepatic phase (every 5–15 min) and immediately post-reperfusion (every 5–10 min) was performed. This frequent sampling revealed that viral clearance is often minimal during the anhepatic phase, which goes against our and others' assumptions in previous models in which similar viral clearance rates were assumed during both the anhepatic and post-reperfusion phases (*Dahari et al., 2005*; *Garcia-Retortillo et al., 2002*; *Hughes et al., 2017*; *Powers et al., 2006*). Specifically, in three of five cases, viral load remained at a plateau in the absence of the liver indicating that not only viral production stopped but that viral clearance stopped as well, suggesting that the liver is involved in the clearance of circulating HCV. Interestingly, however, in Cases 4 and 5, fast viral clearance ($t_{1/2} \sim 1$–2 hr) was estimated via model calibration with measured data during the anhepatic phase while accounting for the recorded high fluid input of 3300 and 3350 mL in Cases 4 and 5, respectively (*Equation. 3*). Theoretically, to explain the fast viral clearance solely based on fluid input, the model (*Equation. 3*) predicts about fourfold higher fluid input (i.e, ~13,000 mL) than the recorded input (not shown), suggesting that other extrahepatic mechanisms must have contributed to viral clearance during the anhepatic phase in these patients.

The current study includes five patients who underwent liver transplant before the era of direct-acting antivirals (DAA) and were viremic at the time of transplant. With the advent of DAA therapy, many patients are treated successfully for HCV before liver transplantation (*Belli et al., 2018*). Nevertheless, the finding of viral steady state during anhepatic phase in the majority of patients and the rapid uptake of HCV by the liver during reperfusion reinforces the benefit of achieving not only viral negativity but also achieving less than one virus particle in the entire extracellular body fluid (i.e., cure boundary recently used in *Etzion et al., 2020*), with DAA therapy prior to transplantation to prevent infection of the graft.

Another novel finding reported herein is that following the introduction and reperfusion of the donor liver, the viral load declined in a biphasic manner (in Cases 1–4) consistent with an extremely

fast viral decline ($t_{1/2}$~ 5 min) that lasted ~14 min post-reperfusion followed by slower decline (HCV $t_{1/2}$= 67 min). To reproduce this biphasic decline, we extended *Powers et al., 2006*, model (*Equation. 3*) by introducing a time-dependent HCV clearance, $c(t)$, as shown in *Equation. 4*. We hypothesize that the initial rapid decrease is due to entrance into and binding of the virus to the new liver reminiscent of the *Ganesan et al., 2011*, findings in a mouse model for adenovirus clearance 1 min after infusion. The slower decrease (i.e., HCV $t_{1/2}$ of 67 min) likely represents a more physiological (i.e., infectious) uptake rate of virions by the new liver before new virions are likely to be released from infected cells. Previous efforts to measure serum HCV virion clearance have been performed in chronically infected patients receiving antiviral drugs that inhibit viral production (*Dahari et al., 2011*). Interestingly, the 67 min for HCV $t_{1/2}$ estimated in the current study is consistent with previous estimates of the HCV $t_{1/2}$ of 45 min in patients treated with a potent inhibitor, daclatasvir, that blocks virus production (*Guedj et al., 2013*).

While in Cases 1–4, a rapid resurgence of HCV was observed 6 hr post-reperfusion, in Case 5, an extremely slow viral decline was observed during the first 4 hr post-reperfusion (*Figure 3* and *Table 3*), followed by a viral plateau (~2.5 log IU/ml) during the subsequent 6 days (*Figure 1A* and *Supplementary file 3*, Table S3). This might be explained by graft dysfunction due to ischemia-reperfusion injury altering HCV binding to its receptors on hepatocytes. We previously reported a slow viral decline post-reperfusion (over 6 hr) in 3 (of 20) patients who underwent liver transplantation, where 2 of the 3 patients had a large degree of ischemia-reperfusion injury (*Garcia-Retortillo et al., 2002*). In addition, closure of porto-collateral vessels is not immediate after transplantation and in some patients blood shunting might contribute to a more modest decrease in HCV viral load during reperfusion (*Navasa et al., 1993*). In addition, we previously observed, in some patients, an extremely slow viral decline post-reperfusion and predicted, via mathematical modeling, the existence of extrahepatic HCV contribution compartment (see *Table 1* in *Dahari et al., 2005*). Thus, the extremely slow viral load decline (or plateau) post-reperfusion in Case 5 might also be explained by the existence of an extrahepatic (or second) replication compartment. Since Case 5 had viral resurgence 3 weeks post-LT (not shown), we cannot rule out that the observed viral plateau of ~2.5 log IU/mL represents viral production from newly infected hepatocytes of the grafted liver with temporary (~3 weeks) extremely limited viral spread.

Two models (*Equation. 3 and 4*) were used in the current study, to separately account for viral clearance during the absence of the liver (i.e., anhepatic phase) and immediately after graft reperfusion (i.e., in the presence of a liver), respectively. When fitting the models to the viral load data, an *individualized* (per case) modeling calibration approach of *Equation. 3 and 4* with the measured viral data was deemed optimal due to the small number ($N$ = 5) of cases for whom very frequent viral sampling was measured during the anhepatic phase and immediately after graft reperfusion, respectively. A *population* mixed-effects (pME) modeling approach was found inappropriate for two main reasons. (i) some of the population probability distributions of the model parameters were not identifiable and it wrongly predicted a viral decline with HCV $t_{1/2}$~98 min (instead of a viral plateau that was not significantly different than slope 0, i.e., very long HCV $t_{1/2}$) during the anhepatic phase in Cases 1–3, if the clearance rate is assumed to have the same median in the population. (ii) Assuming a bimodal distribution for the virus clearance during the anhepatic phase (i.e., different parameter medians for Cases 1–3 and Cases 4 and 5, respectively) reproduced the viral plateau observed on Cases 1–3, but most of the model parameters were no longer identifiable (not shown). In general, the small sample size of $N$ = 5 was insufficient to accurately train a predictive, pME model. Furthermore, it appears unlikely that additional HCV kinetic data can be acquired to explore mixed-effect models since currently patients are treated successfully with DAA before liver transplantation (*Feld et al., 2020*).

In the absence of confounding different cell types, our cell culture experiments support the notion that hepatocytes not only produce the virus, but also may play a role in clearing the virus from the extracellular space even if the cells are already chronically infected. Reminiscent of what was observed in vivo, the $t_{1/2}$ of HCV in vitro was significantly shorter in the presence of Huh7 cells that were pharmacologically inhibited from secreting virus compared to in the absence of cells. We used chronically infected cells to avoid any initial rapid binding and influx of HCV as would be expected if naïve Huh7 cells were introduced and utilized the well-characterized drugs daclatasvir or naringenin at doses known to effectively block de novo virus secretion (*Goldwasser et al., 2011*; *Hu et al., 2020*). Because neither daclatasvir nor naringenin are known to enhance the degradation rate of HCV particles in the

media, the data suggest that a large amount of the virus in the media enters and/or binds to the cells over time. Of note, the HCV $t_{1/2}$ measured in the presence of cells in vitro may be overestimated if daclatasvir or naringenin do not completely block cell culture produced HCV (HCVcc) secretion, hence the difference in HCV $t_{1/2}$ calculated compared to the absence of cells may be underestimated.

While we find this in vitro data interesting, it is important to note that hepatoma cells in vitro and the HCVcc virions they produce are not identical to in vivo infection and thus more investigation is needed before specific conclusions can be drawn directly from the in vitro data. Because the vast majority of HCV RNA in the culture media in vitro is non-infectious (e.g., specific infectivity of 1:500–1:1000) (*Wakita et al., 2005*; *Cai et al., 2005*; *Hueging et al., 2015*; *Sarhan et al., 2017*) and superinfection exclusion has been reported (*Schaller et al., 2007*), we assume this entry is primarily non-productive. However, whether this non-productive uptake involves an HCV-specific entry mechanism (i.e., involves HCV receptors) or represents some sort of general supernatant sampling, by which the hepatoma cells may be acquiring nutrients or clearing debris remains to be determined. This could be investigated experimentally in a variety of ways through the use of different cells (e.g., primary hepatocytes and HCV non-permissive hepatic and non-hepatic cells), different HCV sources (e.g., HCVcc entry defective HCV particles and clinically obtained HCV), or even the use of specific HCV entry inhibitors. However, because it is suspect to extrapolate quantitative differences in cell behavior from cell culture to the in vivo situation, questions regarding the relative contribution of the different cells in the liver to the observed HCV clearance in patients will likely require the use of in vivo models of HCV infection, such as urokinase-type plasminogen activator/severe combined immunodeficiency mice transplanted with human hepatocytes (*Uchida et al., 2017*). The current data simply raises the possibility that hepatocytes might be involved and helps justify the further investigation of this hypothesis.

The notion of the liver playing a significant role in the clearance of pathogens from circulation is not new. Others have suggested hepatic involvement in the clearing of simian immunodeficiency virus and adenovirus in animals (*Ganesan et al., 2011*; *Zhang et al., 2002*). However, to our best knowledge, the current study is the first one to investigate this in vitro, providing evidence that hepatocytes may contribute to viral clearance, but it remains to be determined the extent to which hepatocytes contribute to HCV clearance and if the involvement of hepatocytes might be different for non-hepatotropic viruses.

## Materials and methods
### Patients
Five transplant recipients (termed cases) with median age 60 years (range, 49–68) and median BMI 30 kg/m$^2$ (range, 19–34) underwent liver transplantation (*Table 1*). Four of the five liver donors were male with median age 69 years (range, 29–73) (*Table 1*). All patients received 500 mg methylprednisolone immediately before laparotomy and 500 mg during graft reperfusion. For each patient, blood samples (2 mL) were taken before liver transplantation, every 5–15 min during the anhepatic phase for a total of 7–12 samples, every 5 min during the first hour of reperfusion, and every 10 min until the end of surgery for a total of 15–21 samples. Thereafter, blood was sampled every hour for the first 6 hr, every 6 hr during the first day, and daily during the first week. Transfusion of blood products, saline, and albumin were recorded. The study was approved by the Ethics Committee at Hospital Clinic Barcelona and all patients provided a written informed consent.

### Mathematical modeling of HCV kinetics during liver transplantation
A schematic of the mathematical modeling approach utilized to understand HCV dynamics during the different stages of liver transplantation is shown in *Figure 2*. Shortly before liver removal, we assume that virus is produced at constant rate $P$ and cleared from the serum at rate $c$ per virion as in *Neumann et al., 1998*:

$$dV/dt = P - cV \tag{1}$$

where $V$ is the viral load, expressed in international units (IU) per mL. Assuming the pre-transplant viral load to be at steady state, that is, $dV/dt = 0$, implies that HCV production and clearance balance or $P =$

$cV$. During the anhepatic phase and 4 hr after reperfusion, we assume that in the absence of the liver (i.e., anhepatic phase), virus production ceases, $P = 0$. Hence, **Equation 1** reduces to

$$dV/dt = -c_{AH}V \tag{2}$$

where parameter $c$ in **Equation. 1** is replaced by $c_{AH}$ to represent viral clearance during the anhepatic phase. Some patients received albumin and blood transfusions to compensate for the fluid loss that occurred during transplant surgery. The blood loss causes a reduction in the absolute amount of virus in circulation, but does not affect the virus concentration, $V$. In contrast, the fluid intake results in a dilution of $V$. We can describe the change in patient viral load during the anhepatic phase as in **Powers et al., 2006**.

$$\frac{dV}{dt} = -\left(c_{AH} + \frac{i}{F}\right) V$$
$$\frac{dF}{dt} = i - \lambda \tag{3}$$

where $F$ denotes the total body extracellular fluid which can vary over the course of the anhepatic phase, $i$ is the rate of fluid intake, and $\lambda$ is the rate of blood/fluid loss.

To account for the initial rapid and second slower phases of HCV decline post-reperfusion (RP) in Cases 1–4, we modified **Equation. 3** by including a time-dependent viral clearance rate $c_t(t)$.

$$\frac{dV}{dt} = -\left(c_t(t) + \frac{i}{F}\right) V$$
$$\frac{dF}{dt} = i - \lambda \tag{4}$$
$$c_t(t) = c_0 + (c_{RP} - c_0)(1 - e^{-\kappa t})$$

where $c_0$ represents the initial rapid clearance rate, $c_{RP}$ the second slower clearance rate, and $\kappa$ governs how quickly $c_0$ reduces to $c$. Note that if $\kappa = 0$ or if $\kappa$ is very large then $c_t(t)$ reduces to a constant. The units of $c_{RP}$ and $c_0$ are both 1 /day, $V$ is in IU/mL, $F$ is in mL, $i$ and $\lambda$ are in mL/day, $\kappa$ is in units of 1 /day and $t$ is in days. For clarity parameter, units are sometimes converted to other units (e.g., liter to mL and day to hour).

The fluid input and loss rates $i, \lambda$ were computed by the ratio of recorded volumes of fluid input and loss divided by each patient's duration in the anhepatic phase. In the absence of surgical complications, the fluid loss and intake are usually approximately balanced, though loss is generally slightly more than the input. Since the exact volume of blood loss was not available, we evaluated the maximum impact of fluid intake which occurs when the blood loss and input are balanced, that is, $\lambda = i$. When $\lambda \le i$, the total fluid volume $F$ is a non-decreasing function over time, which attains a minimum when $\lambda = i$. As a consequence, the impact of $i/F$, involved in describing viral concentration dynamics, is maximum if $\lambda = i$.

Several studies have previously reported that HCV can be detected in different human body fluids of HCV-infected patients (**Wang et al., 2006**; **Suzuki et al., 2005**; **Ortiz-Movilla et al., 2002**; **Bourlet et al., 2002**; **Leruez-Ville et al., 2000**; **Mendel et al., 1997**; **Numata et al., 1993**; **Pfaender et al., 2018**). Therefore, the total fluid volume before surgery, $F(0)$, was calculated using the patient's body weight (**Table 1**) under the assumption of 15 L of extracellular fluid per 70 kg. In addition, we explored utilizing just 5 L of blood per 70 kg since transfusions are mainly given to compensate for blood loss during transplantation and HCV viral load is typically higher by several orders of magnitude in the blood compared to the other diverse body fluids (**Numata et al., 1993**; **Pfaender et al., 2018**). Lastly, parameter $i$ was estimated by taking the recorded volumes of fluid input divided by each phase (actual anhepatic and 4 hr post-reperfusion).

Model calibration was performed using Python 3.7.4 together with Scipy Version 1.3.1 and Numpy Version 1.17.2. The fitting used the optimize.least_squares function in Scipy, which minimizes the squares of the differences between the model and patient data (least squares). The minimization is done using the Trust Region Reflective Algorithm 'trf,'. For the linear regression, the *linregress* function was used from the scipy.stats package. Results with p-values $\le 0.05$ were considered statistically significant. The viral slope was considered flat (or plateau) if the estimate was not significantly different from 0, that is, $p > 0.05$. To identify coupled parameters (i.e., identifiability issues), the optimize.least_squares provides the covariance matrix of the resulting fit, which was converted to the correlation

matrix by normalizing the covariances by the standard deviations of the corresponding variables (*Chis et al., 2016*).

## Cells and virus

Huh7 human hepatoma cells (JCRB0403; RRID.CVCL_0336) were obtained from the Japanese Collection of Research Bioresources Cell Bank. Cell stocks have only been passaged 19 times. Every time a new stock is generated, cells are mycoplasma-tested prior to freezing. New aliquots are thawed after 10 passages and cultured in complete Dulbecco's modified Eagle's medium supplemented with 100 units/mL penicillin, 100 mg/mL streptomycin, 2 mM L-glutamine (Corning), 10 mM HEPES (Santa Cruz), and non-essential amino acids (Thermofisher Scientific) and 10 % fetal bovine serum (Hyclone or Gibco) at 37 °C in 5 % $CO_2$. Stocks of HCVcc were generated as previously described from a plasmid encoding the JFH-1 genome that was provided by Takaji Wakita (National Institute of Infectious Diseases, Tokyo, Japan) (*Zhong et al., 2005*; *Yu and Uprichard, 2010*; *Wakita et al., 2005*).

## Measuring HCV half-life in vitro

To measure the HCVcc half-life in the absence of cells, the JFH-1 HCVcc stock was diluted to $10^4$ ffU/mL, which is the level observed in culture media during steady state in the JFH-1 HCVcc infection system (*Zhong et al., 2005*) and incubated at 37 °C simulating conditions during HCV infection in cell culture. At indicated time points, medium was harvested from duplicate wells and frozen at –80 °C until HCV RNA was isolated for quantification.

To measure HCVcc half-life in the presence of chronically HCV-infected cells, confluent non-growing Huh7 cells (~60,000 cells/well) (*Yu and Uprichard, 2010*) were infected with JFH-1 HCVcc at a low MOI (0.01) and cultured for 12 days until HCV levels reached a steady state. Cells were then treated with 1 nM daclatasvir (provided by Bristol-Myers Squibb, NYC, NY) to stop HCV replication and infectious virus secretion (a dose 36 times the reported $EC_{50}$ [19]) or 200 μM naringenin to block HCV secretion (a dose reported to block 74 % of JFH-1 HCVcc secretion; *Goldwasser et al., 2011*). Cultures were immediately placed back at 37 °C and at indicated times post-treatment, medium was harvested from replicate wells (the number of replicate wells included in each experiment is indicated in *Figure 4*). Because the goal was to compare HCV disappearance from the media in the absence and presence of a confluent cell monolayer, and the cells completely covered the bottom of the cell culture-treated 96-well plate in which they were plated, we utilized non-cell cultured treated plastics for the 'no cell' condition.

Mouse liver RNA was added as an internal extraction efficiency control and total RNA was isolated using an ABI PRISM 6100 Nucleic Acid Prepstation (Applied Biosystems), using the manufacturer's instructions. RNA was used for random primed cDNA synthesis using Fermentas reverse transcriptase reagents (ThermoScientific), followed by real-time PCR quantification in technical replicate using an Applied Biosystems 7300 real-time thermocycler (Applied Biosystems). Thermal cycling consisted of an initial denaturation step for 10 min at 95 °C followed by 40 cycles of denaturation (15 s at 95 °C) and annealing/extension (1 min at 60 °C). HCV levels were determined relative to an HCV standard curve and normalized to carrier RNA levels (as a control for recovery efficiency). The PCR primers used to detect HCV were 5'-CGACACTCCACCATAGATCACT-3'_ (sense) and 5'-GAGGCTGCACGACACT CATACT-3'_ (antisense).

## Acknowledgements

The study was supported, in part, by U.S. National Institute of Health grants, R01-AI078881, R01-OD011095, R01GM121600, and R01-AI116868, Instituto de Salud Carlos III (PI15/00151 and PI13/00155,cofunded by the European Regional Development Fund [ERDF]) and by Secretaria d'Universitats i Recerca del Departament d'Economia i Coneixement (grant 2017_SGR_1753) and CERCA Programme/Generalitat de Catalunya, and Germany Academic Exchange Service.

## Additional information

### Competing interests

Xavier Forns: XF acted as advisor for Gilead and AbbVie. The other authors declare that no competing interests exist.

## Funding

| Funder | Grant reference number | Author |
| --- | --- | --- |
| National Institute of Allergy and Infectious Diseases | R01-AI078881 | Susan L Uprichard |
| National Institute of Allergy and Infectious Diseases | R01-AI116868 | Alan S Perelson |
| Instituto de Salud Carlos III | PI15/00151 | Xavier Forns |
| Secretaria d'Universitats i Recerca del Departament d'Economia i Coneixement | grant 2017_SGR_1753 | Xavier Forns |
| CERCA Programme/ Generalitat de Catalunya | | Xavier Forns |
| Germany Academic Exchange Service | | Tje Lin Chung |
| National Institute of General Medical Sciences | R01GM121600 | Harel Dahari |
| Instituto de Salud Carlos III | PI13/00155 | Sofía Pérez-del-Pulgar |
| National Institutes of Health | R01-OD011095 | Alan S Perelson |

The funders had no role in study design, data collection and interpretation, or the decision to submit the work for publication.

## Author contributions

Louis Shekhtman, Formal analysis, Investigation, Methodology, Writing – original draft, Writing – review and editing; Miquel Navasa, Investigation, Methodology, Writing – original draft; Natasha Sansone, Formal analysis, Investigation, Methodology, Writing – review and editing; Gonzalo Crespo, Investigation, Writing – review and editing; Gitanjali Subramanya, Investigation, Methodology, Writing – review and editing; Tje Lin Chung, Investigation, Methodology, Writing – original draft, Writing – review and editing; E Fabian Cardozo-Ojeda, Validation; Sofía Pérez-del-Pulgar, Funding acquisition, Investigation, Writing – review and editing; Alan S Perelson, Funding acquisition, Writing – review and editing; Scott J Cotler, Writing – original draft, Writing – review and editing; Xavier Forns, Conceptualization, Funding acquisition, Supervision, Writing – original draft, Writing – review and editing; Susan L Uprichard, Conceptualization, Funding acquisition, Methodology, Supervision, Writing – original draft, Writing – review and editing; Harel Dahari, Conceptualization, Formal analysis, Investigation, Methodology, Supervision, Writing – original draft, Writing – review and editing, Funding acquisition

## Author ORCIDs

Louis Shekhtman http://orcid.org/0000-0001-5273-8363
Miquel Navasa http://orcid.org/0000-0002-3130-9604
E Fabian Cardozo-Ojeda http://orcid.org/0000-0001-8690-9896
Sofía Pérez-del-Pulgar http://orcid.org/0000-0002-9890-300X
Alan S Perelson http://orcid.org/0000-0002-2455-0002
Xavier Forns http://orcid.org/0000-0002-8188-1764
Susan L Uprichard http://orcid.org/0000-0001-5691-1557
Harel Dahari http://orcid.org/0000-0002-3357-1817

## Ethics

Human subjects: The study was approved by the Ethics Committee at Hospital Clinic Barcelona (Record 2010/5810) and all patients provided a written informed consent.

## Decision letter and Author response

Decision letter https://doi.org/10.7554/eLife.65297.sa1
Author response https://doi.org/10.7554/eLife.65297.sa2

## Additional files

### Supplementary files

• Supplementary file 1. Table S1. Best-fit parameter estimates determined by fitting *Equation. (3)* with data obtained during the anhepatic (AH) phase, assuming extracellular fluid volume of 5 L and that fluid intake and outtake are equal (see Materials and methods). VP, viral plateau (not significantly different from slope 0). Note that the median is not provided because many patients had a best-fit value of 0 for the slope precluding an estimate of a half-life.

• Supplementary file 2. Table S2. Best-fit parameter estimates determined by fitting *Equation. (4)* with data obtained during the 4 hr after graft reperfusion (RP), assuming extracellular fluid volume of 5 L and that fluid intake and outtake are equal (see Materials and methods). VP, viral plateau (not significantly different from slope 0). * Since $c_0$ and $c_{RP}$ were highly correlated and not independently identifiable (correlation matrix) and population modeling (using Monolix) indicated that $c_0$ was not identifiable (not shown), the initial virus clearance rate (i.e., $c_0$ in *Equation. 4*) was fixed to its best-fit value (first estimated with $c_0$, $c_{RP}$, and $\kappa$ as free parameters) and then the errors on the remaining parameters ($c_{RP}$ and $\kappa$) were computed. ** *Equation. 3* was used to estimate hepatitis C virus (HCV) $t_{1/2}$. Since best estimate of clearance was $c_{RP} = 0$, half-life is undefined.

• Supplementary file 3. Table S3. Longer term viral kinetics. We found the time when hepatitis C virus (HCV) VL begins increasing and then determined the half-life of HCV up until the time of the first increase. We then measured the continued decrease from the end of reperfusion (RP) until this time and determined the slope using linear regression. We converted this slope to a half-life and found that the half-life tends to be on the order of several hours (except for Case 5 in whom a viral plateau was observed until the end of the follow-up period).

• Transparent reporting form

### Data availability

All data generated or analyzed during this study are included in the manuscript and supporting files.

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
