## [Editor Report]

In their work, the authors combine clinical data and mathematical modelling to shed light on the role of hepatocytes in HCV clearance. This manuscript will be of interest to clinicians in organ transplantation centers and to translational hepatitis virus researchers given that it provides a rare and carefully collected dataset of hepatitis C virus blood titers during and after liver transplantation. The manuscript is also of potential interest to modelers interested in HCV infection and more broadly infectious disease specialists.

---

## [Decision Letter]

Thank you for submitting your article “Modeling hepatitis C virus kinetics in vivo and in vitro reveals the role of hepatocytes in virus clearance“ for consideration by *eLife*. Your article has been reviewed by 3 peer reviewers, including Niel Hens as the Reviewing Editor and Reviewer #1, and the evaluation has been overseen by Päivi Ojala as the Senior Editor. The following individual involved in review of your submission has agreed to reveal their identity: Melanie Prague (Reviewer #3).

The reviewers have discussed the reviews with one another and the Reviewing Editor has drafted this decision to help you prepare a revised submission.

As the editors have judged that your manuscript is of interest, but as described below that additional experiments are required before it is published, we would like to draw your attention to changes in our revision policy that we have made in response to COVID-19 (https://elifesciences.org/articles/57162). First, because many researchers have temporarily lost access to the labs, we will give authors as much time as they need to submit revised manuscripts. We are also offering, if you choose, to post the manuscript to bioRxiv (if it is not already there) along with this decision letter and a formal designation that the manuscript is “in revision at *eLife*“. Please let us know if you would like to pursue this option. (If your work is more suitable for medRxiv, you will need to post the preprint yourself, as the mechanisms for us to do so are still in development.)

Summary:

Using mathematical modeling applied to a precise collection of blood samples in 5 patients, Shekhtman et al. evaluate the kinetics of hepatitis C viral load during and after liver transplantation. The data suggest that anhepatic (absence of liver) and early post-reperfusion phases of liver transplantation do not have similar HCV clearance rates, and therefore that the liver plays a major role not only in HCV production but also in HCV clearance. It shows indeed, in a fraction of the patients, maintenance of unchanged viral load during the anhepatic phase followed by a prompt decrease of HCV RNA titers upon reperfusion of the new liver graft. This drop could not be attributed to the transfusion of blood products and dilution of the virus load. In fact, the authors report a biphasic decrease of the viral load early after liver reperfusion, with a first abrupt drop in viremia in the first 20 minutes, attributed to HCV uptake in the new liver, and a subsequent slower decline in viral titers in the next 6 hours, suggested to correspond to the physiological HCV clearance rate in absence of new virus production, and in adequation with previous estimates.

This study consolidates previous reports on HCV viremia during liver transplantation, with a limited number of patients (5) but more extensive and precise sampling. The hepatic clearance of HCV is not completely surprising given the high vascularization of the liver. Also, as the authors point out, similar findings were reported by Ganesan et al. for adenovirus clearance, although in this latter study the liver sinusoidal endothelial cells were involved. The study emphasizes the need to clear HCV before performing a liver transplantation in order to prevent the infection of the liver graft. The conceptual advance is limited by the lack of mechanistic study but the authors share a unique dataset. Furthermore, some parts of the manuscripts need to be clarified and the in vitro aspect of the study needs to be better controlled and deepened to support the authors' claims.

Essential revisions:

1) Mathematical model.

Motivation: A clear motivation for the use of the mathematical model as is presented now is missing. The logical flow is there but how did the authors arrive to this model and not to another model as such. The authors should also highlight the novelty of their model as compared to previously published models on the same question. The mathematical modelling resembles in some ways the models by Powers K et al. (PMID 16447184) and Neumann et al. (PMID 9756471). Adequate references should be included and novelties in the proposed model highlighted. They could also comment on the use of the complete body fluid volume rather than the blood volume in their mathematical model; it is not clear for the non-initiated reader whether the HCV titer is the same in the different body fluids and whether all are affected in the same way by the transfusions and blood losses during surgery.

Description: The equation used could be made more accessible for a broad range of readers by introducing more thoroughly the variables (e.g. please introduce V already for equation 1) and including the dimensional analysis for each equation (e.g. [L][T]-1, etc. do c and P correspond to HCV RNA concentrations produced per unit of time?). Please provide a clear description of how t_1/2_ is derived from Equations 3-4.

Statistical methodology:

a. The description of statistical methods for calibration and fitting is very poor. P5 last paragraph. How do parameters in Table 1-3 relate to equation 2-3-4? Which ”regression models“ did the author fit? Authors also mentioned that they fitted model from equation 4, please provide the method. Is it simple least mean square? Why did the author did not adopt a population fitting approach such as described in Guedj et al. 2013 (https://doi.org/10.1073/pnas.1203110110) using (for example) the Monolix software (R package saemix could also be relevant)?

b. In the result section, I am confused how section 1, 2 and 3 relate? Can you explain better what is the added value of ”Viral kinetics before and during the anhepatic phase“ and ”Viral kinetics after graft reperfusion“, when a clearer/deeper description of the results in the “Modeling HCV kinetics“ would probably carry the same information? Please clarify the analysis done (as e.g. fitting individual data) and present the parameter values (effect sizes in Table 2; please do not use a median based on decreasing slopes only) etc. To what extent is the model well specified and parameters identifiable? It is disturbing to change the model of analysis because of data (see example of patient 5) – i.e. conditioning the model structure on data observation. If the model is flexible enough, it should be possible to fit the data assuming some parameters such as c very large. Being able to keep the same dynamical model for all patients would help warranting its validity. I strongly believe that analyzing jointly all the observed course of trajectories (before AH, during AH and after RP) will have a strong added value.

2) The in vitro evidence is weak and lack controls. Because it relies on a hepatoma cell line rather than primary cells and does not include any other cell type as a comparison the depicted assay does not support a larger role of hepatocytes as compared to other liver or non-liver cell types in HCV uptake. Furthermore, controls are missing that should show the complete block in viral RNA secretion in this system. In particular, liver sinusoidal endothelial cells were proposed by Ganesan and colleagues as playing a major role in virus uptake in the liver, and the role of this cell type should be tested if the authors want to make a claim on the cell type responsible for the post-transfusion decline in HCV titers. Also, there seems to be a confusion between technical and biological replicates in this in vitro data.

The cell culture experiment does not seem valid to test the author´s hypothesis and more specifically quantify the role of hepatocytes, as indicated by the authors. Additional controls and details would be instructive.

a. Are the hepatoma cells confluent (can one exclude virus adsorption to the dish)? Is the control condition (absence of cells, Figure 4A) in a tube or a cell culture-coated dish?

b. Why using HCV-infected cells in Figure 4B? The authors mentioned they want to avoid any initial rapid binding and influx of HCV as expected if using Huh7 cells, but wouldn´t this in fact mimic the situation of the reperfusion of a naive liver graft? Why not incubating the virus stock used in A on different (non-infected) cell culture dishes: empty dish (control for virus / viral RNA adsorption to plastic dish) vs. dish containing confluent monolayers of different cell types (different hepatocyte-derived cells, other liver cell types, non-liver cells…). This would be much easier to interpret than the proposed assay, where it is not clear whether virus production is directly abrogated at t=0 post-treatment. The authors should test other liver cells and liver-unrelated cells since they propose that “hepatocytes in particular“ play a major role in circulating HCV clearance. Ideally, the authors should test primary human hepatocytes and LSECs in parallel.

c. The authors should verify that their inhibitors indeed completely block HCV RNA secretion in the conditions tested (for instance by performing a medium change in similarly infected wells at t=0, after which the virus titer in the supernatant should remain at 0 if HCV RNA secretion is blocked). This is unlikely the case, at least for Naringerin: according to Goldwasser et al. (PMID 21354229), 200 μm Naringerin merely decreased 4 fold HCV RNA secretion.

3) Other essential revisions:

– Blood transfusion, which is an important confounder in the in vivo study, is not sufficiently described and discussed. Can you comment on the relevance of accounting for other covariates (from Table 1 put also possibly the length of HCV infection, type of treatments…). I bet there may be a lack of power in the study but a discussion of possible confounder could be added.

– The authors should discuss the potential role of extrahepatic HCV reservoirs in their study.

– Magnitude of results between in vivo and in vitro study should be better described and compared.

– Can you clarify why you did not model phase D i.e. >4h post RP, see Figure 2?

[Editors' note: further revisions were suggested prior to acceptance, as described below.]

Thank you for resubmitting your work entitled “Modeling hepatitis C virus kinetics during liver transplantation reveals the role of the liver in virus clearance“ for further consideration by *eLife*. Your revised article has been evaluated by Päivi Ojala (Senior Editor) and a Reviewing Editor.

The manuscript has been improved but there are some remaining issues that need to be addressed, as outlined below:

Both reviewers have acknowledged the authors' efforts in providing a revision of their work addressing most of their comments raised. There is, however, some more work needed to clarify some outstanding issues.

*Reviewer #2:*

The validity of the approach still suffers from several pitfalls:

1. HCV cell culture systems are mostly limited to cancer cell lines and a particular HCV genotype, hence justifying the approach used by the authors. These systems have proven very useful to dissect HCV replication cycle, however, it is not clear whether they are helpful to support the specific authors conclusions. In particular, the virus genotype, lipoprotein coat and most importantly specific infectivity is different in the cell culture system and might affect HCV genome stability and uptake. Indeed, the authors quantify HCV RNA as a readout for HCV half time. This is coherent with the in vivo readout (where infectivity is difficult to assess) but this RNA might be in completely different forms in JFH1 HCVcc as compared to infected patient serum. In fact, Lindenbach and colleagues, PNAS 2006, reported a 100x lower specific infectivity for cell culture virus as compared to virus retrieved from animals. The relative proportions of subgenomic, naked, encapsidated, enveloped, lipoviroparticle-associated, exosome-associated viral RNA might be completely different from what is found in patient serum samples. On the other hand, the cancer cell lines in culture might behave differently than hepatocytes in clearing HCV genome, whether by uptake or RNase degradation inside or outside the cells. Would any other cancer cell line give the same clearance effect? If yes, how does it support the authors’ point: does the liver clear many viruses simply because it is very vascularized?

2. I agree that the definition of biological replicates is problematic and subject to interpretation in particular with cell lines. However, as stated in the authors’ response, biological replicates refer to “biologically distinct samples“. If I understood well, the “biological duplicates“ described in Figure 4 are replicate wells of the Huh7 cell line that were seeded, treated and infected in parallel, with the same cell passage, virus and inhibitor stocks, on the same day, which to me does not quality as biologically distinct samples, and the variation observed between these wells is mostly technical (as opposed to different mice in an animal experiment for instance). I therefore recommend removing “biological duplicate“ from the figure legend. Since the data shown is representative of 2 experiments, the conclusion would be stronger by averaging the 2 independent experiments.

Given the low impact of the cell culture experiment on the authors conclusions and its strong limitations, I recommend the authors significantly reinforce the in vitro evidence and discuss the remaining limitations more in detail (as discussed above and in the previous review: testing other cell types, checking infectivity in addition to genome copies, etc).

*Reviewer #3:*

Figure 1 could be made of 2 figures: 1/ the existing one 2/ a zoom in in the first few hours.

---

## [Author Response]

Essential revisions:1) Mathematical model.Motivation: A clear motivation for the use of the mathematical model as is presented now is missing. The logical flow is there but how did the authors arrive to this model and not to another model as such. The authors should also highlight the novelty of their model as compared to previously published models on the same question. The mathematical modelling resembles in some ways the models by Powers K et al. (PMID 16447184) and Neumann et al. (PMID 9756471). Adequate references should be included and novelties in the proposed model highlighted. They could also comment on the use of the complete body fluid volume rather than the blood volume in their mathematical model; it is not clear for the non-initiated reader whether the HCV titer is the same in the different body fluids and whether all are affected in the same way by the transfusions and blood losses during surgery.

We have cited Neumann et al. before introducing Equation 2 in the revised manuscript (Line 217). We also note that the reviewer may have overlooked that Powers et al. was already cited in the original submission. To enhance clarity, we add “Powers et al” before its citation in the revised manuscript (Line 226).

Regarding the choices in the model, we have expanded our model description to better explain and justify some of the modelling choices (Lines 222, 233-235, 243-249). We also make it clear what is novel about Equation 4 in the revised manuscript (see the paragraph in the Discussion that begins with “Another novel finding…”).

In addition, the referee makes an important point about the use of total extracellular body fluid volume compared to the blood volume and the issue of HCV titers in different body fluids. Several studies have previously reported that HCV RNA can be detected in different human body fluids of HCV-infected patients (refs 7-14 in the revised manuscript), supporting our decision to account for the entire extracellular body fluid volume in our model (Lines243-248). Since transfusions are mainly given to compensate for blood loss and HCV viral load is typically higher by several orders of magnitude in the blood compared to the other body fluids, we also included in the revised manuscript a supplementary Table S1 to show that using just blood volume of 5L (compared to 15L of entire extracellular body fluid volume including blood) did not significantly affect the results of the study. We made it clear in Methods of the revised manuscript about the studies showing that HCV is present in various body fluids of infected patients and note the additional supplementary modeling results using 5L of body fluid volume (Lines 75 and 85).

Description: The equation used could be made more accessible for a broad range of readers by introducing more thoroughly the variables (e.g. please introduce V already for equation 1) and including the dimensional analysis for each equation (e.g. [L][T]-1, etc. do c and P correspond to HCV RNA concentrations produced per unit of time?). Please provide a clear description of how t_1/2_ is derived from Equations 3-4.

We have included descriptions of variables immediately where they appear (such as V in Equation 1) and added a discussion of the dimensions of each variable (Lines 222, 238-240). Lastly, we added equations in Tables 4 and 5 showing how HCV t_1/2_ was derived from the HCV clearance parameters, C_AH_, C_0_ and C_RP_ in Equations (3)-(4). We also for clarity renamed viral clearance (c) during AH as c_AH_ and during RP as c_RP_.

Statistical methodology:a. The description of statistical methods for calibration and fitting is very poor. P5 last paragraph. How do parameters in Table 1-3 relate to equation 2-3-4? Which “regression models” did the author fit? Authors also mentioned that they fitted model from equation 4, please provide the method. Is it simple least mean square? Why did the author did not adopt a population fitting approach such as described in Guedj et al. 2013 (https://doi.org/10.1073/pnas.1203110110) using (for example) the Monolix software (R package saemix could also be relevant)?

We have expanded significantly on our model calibration and fitting approach description in the revised manuscript (Lines 256-258). All models were indeed fit using least squares and with the Trust Region Reflective Algorithm (‘trf’) as implemented in Scipy.

We also described in the Results sections on viral kinetics (Lines 55, 64-65), in Methods (Lines 258-9), and in table legends (Lines 411 and 426) that Tables 2-3 are the results of linear regression. The modelling section now clearly emphasizes that only there is the modelling applied, whereas the kinetics section only using linear regression.

Because population modeling approaches are not designed for studies with small N (N<~8), we did not originally include this approach. However, we went ahead and explored it as the reviewer suggested and confirmed that it is not suitable for the current study. We added a new paragraph in the Discussion of the revised manuscript about this (Lines 159-174).

The following results (Author response table 1; Author response table 2 and Author response table 3) are from Monolix after assuming similar probability distribution for all model parameters among all 5 Cases wrongly predicted a viral decline (instead of a plateau) during the anhepatic phase in Cases 1-3.

**Author response table 1. sa2table1:** Maximum likelihood estimates of the model parameters using mixed effects modeling in the software Monolix. The mixed-effects model assumes a population probability distribution with form: log⁡x∼N(log⁡x^,ωx2) with x representing parameters κ, cah, crp and c0, and a population probability distribution: x∼N(x^,ωx2) for parameter  log10⁡V0. Parameters cah and crp represent the virus clearance rate c during the anhepatic phase and the final virus clearance rate after liver graft reperfusion, respectively. %RSE is the residual standard error and quantifies the identifiability of a parameter with the available data—%RSE greater than 50% means that the parameter is not identifiable. With these parameters the estimated median of the HCV t_1/2_ during the anhepatic phase and after liver graft reperfusion are 99 and 101 minutes, respectively, and the initial HCV t_1/2_ right after the anhepatic phase is 0.004 minutes. These estimations are based assuming the total fluid volume before surgery F(0)=1000mlL×15L70Kg×weight (kg), having F units of mL.

Mixed effects model	Parameter	Units	Value	%RSE*
Fixed effects(Population value, x^)	κ^	1/hr	5.13	25.3
	c^ah	1/hr	0.42	23
	c^rp	1/hr	0.41	33.2
	c^0	1/hr	10337.1	235*
	log10⁡V0	Log10 IU/mL	5.6	8.29
Standard deviation of the random effects(variability in the population, ωx)	ωκ	1/hr	0.15	132*
	ωcah	1/hr	0.29	85.6*
	ωcrp	1/hr	0.69	35.6
	ω0	1/hr	1.65	90.6*
	ωlog10⁡V0	Log10 IU/mL	1.04	31.7

**Author response table 2. sa2table2:** Individual parameter estimates using Monolix. These are the parameters used for the model fits in Author response image 1.

id	κ[1/hr]	cah[1/hr]	crp[1/hr]	c0[1/hr]	log10⁡V0[Log10 IU/mL]
p1	4.59	0.39	0.51	33095.23	6.42
p2	5.62	0.48	0.72	3997.18	6.15
p3	5.29	0.4	0.44	18529.58	5.47
p4	4.73	0.57	0.69	28783.43	6.34
p5	5.65	0.34	0.13	3281.26	3.65

**Author response table 3. sa2table3:** Individual estimates of HCV half-life using values from Author response table 2.

id	κ1/days	HCV t_1/2_ [min]during anhepatic phase	HCV t_1/2_ [min] after liver graft reperfusion	Init. HCV t_1/2_ [min]after liver graft reperfusion
p1	110.16	106.64	81.55	0.001
p2	134.88	86.64	57.76	0.010
p3	126.96	103.97	94.52	0.002
p4	113.52	72.96	60.27	0.001
p5	135.60	122.32	319.91	0.013

**Author response image 1. sa2fig1:** Model fits from Monolix using parameters from Author response table 2.

*, Residual standard error, RSE, higher than 50% is considered not identifiable.

The following results (Author response image 2; Author response table 4 and Author response table 5; Author response table 6) are after assuming a bimodal distribution for the virus clearance during the anhepatic phase (i.e., different parameter medians for Cases 1-3, and Cases 4 and 5, respectively) reproduced the viral plateau observed on Cases 1-3, but most of the model parameters were no longer identifiable (%RSE>50, see Author response table 4):

**Author response image 2. sa2fig2:** Model fits from Monolix using parameters from Author response table 4.

**Author response table 4. sa2table4:** Maximum likelihood estimates of the model parameters using mixed effects modeling in the software Monolix. The mixed-effects model assumes a population probability distribution with form: log⁡x∼N(log⁡x^,ωx2) with x representing parameters κ, crp and c0, and a population probability distribution: x∼N(x^,ωx2) for parameter  log10⁡V0. Parameter cah had distribution log⁡cah∼N(log⁡c^ah+β,ωx2) with β=0 for patients 1-3 (i.e. c^ahp1−3 =c^ah) and β>0 for patients 4-5 (i.e. c^ahp4−5 =c^aheβ). Parameters cah and crp represent the virus clearance rate *c* during the anhepatic phase and the final virus clearance rate after liver graft reperfusion, respectively. As before, %RSE greater than 50% means that the parameter is not identifiable. With these parameters the estimated median of the HCV t_1/2_ during the anhepatic phase are 16 hours and 98 minutes for patients 1-3 and patients 4-5, respectively (not enough data to test significance). The median of the HCV t_1/2_ after liver graft reperfusion was 99 minutes, respectively, and the initial HCV t_1/2_ right after the anhepatic phase is 0.004 minutes. These estimations are based assuming the total fluid volume before surgery F(0)=1000mlL×15L70Kg×weight (kg), having F units of mL.

Mixed effects model	Parameter	Units	Value	%RSE*
Fixed effects(Population value, x^)	κ^	1/hr	6.41	22.6
	c^ahp1−3	1/hr	0.043	309*
	c^ahp4−5	1/hr	0.42	136*
	c^rp	1/hr	0.42	32.4
	c^0	1/hr	106223.9	238*
	log10⁡V0	Log10 IU/mL	5.54	8.29
Standard deviation of the random effects(variability in the population, ωx)	ωκ	1/hr	0.17	78.9*
	ωcah	1/hr	0.5	69.5*
	ωcrp	1/hr	0.67	35.5
	ω0	1/hr	1.06	201*
	ωlog10⁡V0	Log10 IU/mL	1.02	31.7

**Author response table 5. sa2table5:** Individual parameter estimates using Monolix. These are the parameters used for the model fits in Author response image 2.

id	κ[1/hr]	cah[1/hr]	crp[1/hr]	c0[1/hr]	log10⁡V0[Log10 IU/mL]
p1	5.47	0.046	0.52	214764.41	6.31
p2	7.42	0.047	0.72	74381.69	6.05
p3	6.47	0.045	0.44	155523.87	5.34
p4	5.64	0.63	0.71	167066.1	6.36
p5	7.3	0.3	0.14	70417.29	3.63

**Author response table 6. sa2table6:** Individual estimates of HCV half-life using values from Author response table 5.

id	κ1/days	HCV t_1/2_ [min]during anhepatic phase	HCV t_1/2_ [min] after liver graft reperfusion	Init. HCV t_1/2_ [min]after liver graft reperfusion
p1	131.28	904.11	79.98	0.0002
p2	178.08	884.87	57.76	0.0006
p3	155.28	924.20	94.52	0.0003
p4	135.36	66.01	58.58	0.0002
p5	175.20	138.63	297.06	0.0006

*, Residual standard error, RSE, higher than 50% is considered not identifiable.

b. In the result section, I am confused how section 1, 2 and 3 relate? Can you explain better what is the added value of “Viral kinetics before and during the anhepatic phase“ and ”Viral kinetics after graft reperfusion“, when a clearer/deeper description of the results in the “Modeling HCV kinetics“ would probably carry the same information? Please clarify the analysis done (as e.g. fitting individual data) and present the parameter values (effect sizes in Table 2; please do not use a median based on decreasing slopes only) etc. To what extent is the model well specified and parameters identifiable? It is disturbing to change the model of analysis because of data (see example of patient 5) – i.e. conditioning the model structure on data observation. If the model is flexible enough, it should be possible to fit the data assuming some parameters such as c very large. Being able to keep the same dynamical model for all patients would help warranting its validity. I strongly believe that analyzing jointly all the observed course of trajectories (before AH, during AH and after RP) will have a strong added value.

We agree that an alternative structure for the paper could incorporate the modelling results immediately after the model description, however we believe that among the journal’s interdisciplinary audience there are likely many non-modeler readers who prefer a separate Results section enabling them to understand the key biological findings of the paper and who will only briefly review the modeling details in the methods section. Such structures of a separate methods and modeling section are common in cross-disciplinary publications, and we chose to continue that pattern as we believe it is most effective approach for broad audiences. Lastly, we have been requested by *eLife*’s production editor to locate the Methods section in the end of the manuscript (after the Discussion section).

Regarding the issue of HCV kinetics during the anhepatic phase, we have revised Table 2 to include the decrease in slope for all patients regardless of whether that decrease was statistically significant.

Regarding the modeling choices for Case (patient) 5, the decision to revise the model for that patient was because that Case 5 does not experience a second viral decline slope post-reperfusion phase, which is why the two different slopes are needed for the other 4 cases. Since this patient only has a single slope after graft reperfusion, there would indeed be identifiability issues with that Case if the same biphasic model were used, e.g., the first slope could be fixed and parameter kappa could just be very large, or the second slope could simply be equal to the first slope. We agree that ideally the model would not need to be altered in this way, however, Case 5 is unique in several ways as the HCV RNA measurements for this Case were lower to begin with and primarily only decreased during the anhepatic phase itself. The reason for this is not clear.

Lastly, regarding a single model for all trajectories, we would like to note that the model during the various different periods Equations (1)-(4) all have many of the same parameters and the same basic structure of clearance. While we appreciate the referee’s suggestion to combine these into a single model, we feel this is impractical as the model must incorporate the discrete nature of the transplantation process and thereby will have to have some discontinuities (at the times where the liver is removed and then reattached). Likewise, the patient receives fluid input and loses fluid during the anhepatic and graft reperfusion phases in different amounts. We noted in the revised manuscript that we use two models that were separately calibrated with data measured during the anhepatic phase and graft reperfusion in a new paragraph in the Discussion that begins with the words “Two models (Equations 3 and 4) were used……” (Lines 159-174).

2) The in vitro evidence is weak and lack controls. Because it relies on a hepatoma cell line rather than primary cells and does not include any other cell type as a comparison the depicted assay does not support a larger role of hepatocytes as compared to other liver or non-liver cell types in HCV uptake.

The Reviewer is correct that the assay performed does not speak to a *larger* role of hepatocytes as compared to other cell types as our goal was not to compare the relative contributions, but to only support the possible contribution of hepatocytes. We independently observed this phenomenon in our in vitro experiments and felt that including this intriguing parallel data enhances the interest of this primarily in vivo study as it suggests a novel clearance mechanism not previously considered and opens the door to further detailed investigation (Line 110).

Furthermore, controls are missing that should show the complete block in viral RNA secretion in this system.

Because the two inhibitors utilized are extremely well characterized and their inhibition of JFH-1 HCVcc infection in vitro had been quantitatively evaluated many times, we did not repeat that analysis in this publication. That being said, we did fail to provide the necessary references documenting this pre-existing data and have now provided the appropriate references (references 19 and 20 in the current version). Additionally, we have added a sentence to the discussion highlighting the important fact that if these inhibitors did not block secretion completely, that the difference in ½ life between the cell/no cell condition would be even greater (Line 180-182). Hence, the lack of complete inhibition would not reduce the effect we have observed, but rather enhance it.

In particular, liver sinusoidal endothelial cells were proposed by Ganesan and colleagues as playing a major role in virus uptake in the liver, and the role of this cell type should be tested if the authors want to make a claim on the cell type responsible for the post-transfusion decline in HCV titers.

Indeed, if we were trying to make a claim regarding the relative role of hepatocytes verses other cells within the liver, then it would be important to test the various cell types in parallel. However, as noted above, that was not our goal.

Additionally, thinking about how one might go about trying to quantify the relative contributions, we would be hesitant to extrapolate quantitatively from in vitro cell culture conditions to the in vivo situation as different cell types do not necessarily maintain similar levels on in vivo function in vitro. We have done our best to go through the manuscript and make sure we have not suggested that hepatocytes contribute to liver uptake more or less than cell types.

Also, there seems to be a confusion between technical and biological replicates in this in vitro data.

Respectfully, we have utilized the standardly accepted definitions articulated by the NIH:

– Biological replicates are parallel measurements of biologically distinct samples

that capture random biological variation, which may itself be a subject of study or a source of noise.

– Technical replicates are repeated measurements of the same sample that represent independent measures of the random noise associated with protocols or equipment.

By this NIH definition the duplicate wells in the experiment (previously mentioned on page 6 and noted by the reviewer in the comment above) are biological replicates as stated by the NIH, i.e. parallel measurements of biologically distinct samples. These biological replicates were then measured in technical replicate by RT-qPCR (i.e. repeat measurements of the same sample). As is standard, we then simply state how many times a particular experiment was repeated.

The cell culture experiment does not seem valid to test the author´s hypothesis and more specifically quantify the role of hepatocytes, as indicated by the authors. Additional controls and details would be instructive.

As noted above, our goal was *not* to compare the relative contribution, but to only support the possible contribution of hepatocytes. Again, we have made sure that no suggestion of this is contained within the manuscript.

a. Are the hepatoma cells confluent (can one exclude virus adsorption to the dish)? Is the control condition (absence of cells, Figure 4A) in a tube or a cell culture-coated dish?

These non-growing cells were completely confluent. At ~60,000 cells per well, the cells are packed in with absolutely none of the 96 well plate bottom exposed. As such, we utilized non-cell culture treated plastics for the no cell condition to avoid the virus binding to bottom of the wells. We have added this information to the M&M (Lines 277-278).

b. Why using HCV-infected cells in Figure 4B? The authors mentioned they want to avoid any initial rapid binding and influx of HCV as expected if using Huh7 cells, but wouldn´t this in fact mimic the situation of the reperfusion of a naive liver graft?

Our intent was not to try and mimic the in vivo situation, but rather to specifically determine if our hepatoma cells exhibited an alternative HCV uptake aside from the known infectious virus uptake observed by uninfected cells. We know that if we put virus on uninfected Huh7 cells, that it will be taken up. It has also been published that the Huh7-based HCV infection system exhibits superinfection exclusion (PMID: 17287280, 17301154). Hence, we designed the experiment to specifically test if the presence of infected Huh7 cells would result in clearance of HCV from the media.

That being said, it is also worth pointing out that to the extent that we might want to mimic the in vivo condition… it is not the post-reperfusion phase we would try to recapitulate. The more informative situation to mimic, which our in vitro experimental design does do to some extent, is the comparison between the PRE-transplant condition vs the anhepatic phase. What is most telling in the in vivo experiment is that pre-transplant HCV levels are at steady-state indicative of balanced viral production and viral clearance and that surprisingly once the liver is removed, that HCV levels remain at steady state in the majority of patients even though the main source of viral production has been removed (it is this transition that indicates that the liver is also involved in clearance from the serum). in vitro, we have shown many times that we achieve a steady state chronic infection. Because we can control the in vitro situation more precisely, we not only “removed the producer cells completely analogous to the anhepatic phase (by incubating the steady state HCV containing media in the absence of cells), but also additionally blocked virus production while leaving the cells present (by utilizing inhibitors to block virus production). This allowed us to directly compare the intrinsic ½ life of the virus (i.e., in the absence of cells) to the ½ life measured in the presence of the cells not actively secreting virus.

Of note, we do mention in the Discussion that the ½ life of HCV measured in chronically infected patients treated with DSV is ~45 minutes, which not surprisingly is much faster than the 14 hour ½ life measured in vitro which lacks all the other cells and blood components present in vivo, including anti-HCV antibodies.

Why not incubating the virus stock used in A on different (non-infected) cell culture dishes: empty dish (control for virus / viral RNA adsorption to plastic dish) vs. dish containing confluent monolayers of different cell types (different hepatocyte-derived cells, other liver cell types, non-liver cells…). This would be much easier to interpret than the proposed assay, where it is not clear whether virus production is directly abrogated at t=0 post-treatment. The authors should test other liver cells and liver-unrelated cells since they propose that “hepatocytes in particular“ play a major role in circulating HCV clearance. Ideally, the authors should test primary human hepatocytes and LSECs in parallel.

All of the above, is repetition of the points above and has been answered. Importantly, we have removed the phrase “hepatocytes in particular” and “particularly hepatocytes.” This was just intended to emphasize that our data suggested hepatocytes may be involved, not to suggest exclusivity and so we removed the problematic language. Accordingly, we also edited the title of the revised manuscript and noted of future research needed in Lines 203-205.

c. The authors should verify that their inhibitors indeed completely block HCV RNA secretion in the conditions tested (for instance by performing a medium change in similarly infected wells at t=0, after which the virus titer in the supernatant should remain at 0 if HCV RNA secretion is blocked). This is unlikely the case, at least for Naringerin: according to Goldwasser et al. (PMID 21354229), 200 μm Naringerin merely decreased 4 fold HCV RNA secretion.

As the reviewer points out, the effect of these inhibitors is established in the literature. As noted above, we have now provided references for this and indicate in the manuscript that any residual secretion occurring would only serve to make the measure ½ life of the virus longer in the cell-containing situation (19 and 20 in the revised manuscript). As such, the expected lack of complete inhibition would reduce the effect we are observing meaning that we are measuring the minimal possible effect of the cells (Lines 185-187).

3) Other essential revisions:– Blood transfusion, which is an important confounder in the in vivo study, is not sufficiently described and discussed. Can you comment on the relevance of accounting for other covariates (from Table 1 put also possibly the length of HCV infection, type of treatments…). I bet there may be a lack of power in the study but a discussion of possible confounder could be added.

To clarify the issues about blood transfusion in the model, we increased our discussion of the parameters *i* and *F,* which are both related to this important issue (Lines 253-254 and 62-63). We have also shifted the raw values of the blood input to revised Table 1 and then provided in the revised Tables 4-5 the actual values of parameter *i.* We also included in the revised Tables 4-5 the value of *F*, which is determined based on the patient weight. Regarding treatment, we mentioned in the Methods of the original manuscript (Lines 210-211 in the revised manuscript) about the treatment with 500 mg methylprednisolone immediately before laparotomy and 500 mg during graft reperfusion. In relation to the length of chronic HCV infection, it is difficult to know as these individuals reached the point of needing transplantation, suggesting that the length of infection could be on the order of many years. We certainly also agree about a lack of power in the study, however we have very frequent sampling enabling us to better understand what occurs in each phase compared to previous studies noted in the manuscript (Lines 38-39). We discussed challenges (or confounders) and limitations of the small but unique kinetic data in Lines 172-175.

– The authors should discuss the potential role of extrahepatic HCV reservoirs in their study.

Indeed, we previously observed, in some patients, an extremely slow viral slope post reperfusion and predicted, via mathematical modeling, the existence of extrahepatic HCV contribution compartment (ref 1 in the revised manuscript). While we cannot rule out that the extremely slow viral load slope post reperfusion in Case 5 represents the predicted extrahepatic (or second) replication compartment, it seems unlikely since both during the 4 hr post reperfusion and beyond (until viral resurged, as shown in the revised Figure 1) HCV t_1/2_ was on the order of hours (see a new Table S3), whereas the extrahepatic clearance in our previous report was on the order of days (see Table 1 in aforementioned ref 1 ). This was added to the Discussion of the revised manuscript (Lines 152-158).

– Magnitude of results between in vivo and in vitro study should be better described and compared.

As noted above, we would not presume to compare the effects quantitatively between the in vitro and in vivo situation which are different in multiple ways. The in vitro experiment here was provided simply to make the point that it is possible that hepatocytes contribute to the clearance observed in vivo. It is intended a conceptual comparison.

– Can you clarify why you did not model phase D i.e. >4h post RP, see Figure 2?

The focus of this paper was primarily on a high-resolution analysis of the early time during and immediately after reperfusion. Many previous studies such as Dahari et al. (Ref 1) and Powers et al. (ref 5) have modeled and analyzed the long-term dynamics post reperfusion, thus we did not feel the need to go in depth on this point.

[Editors' note: further revisions were suggested prior to acceptance, as described below.]

The manuscript has been improved but there are some remaining issues that need to be addressed, as outlined below:Both reviewers have acknowledged the authors' efforts in providing a revision of their work addressing most of their comments raised. There is, however, some more work needed to clarify some outstanding issues.Reviewer #2:The validity of the approach still suffers from several pitfalls:1. HCV cell culture systems are mostly limited to cancer cell lines and a particular HCV genotype, hence justifying the approach used by the authors. These systems have proven very useful to dissect HCV replication cycle, however, it is not clear whether they are helpful to support the specific authors conclusions. In particular, the virus genotype, lipoprotein coat and most importantly specific infectivity is different in the cell culture system and might affect HCV genome stability and uptake. Indeed, the authors quantify HCV RNA as a readout for HCV half time. This is coherent with the in vivo readout (where infectivity is difficult to assess) but this RNA might be in completely different forms in JFH1 HCVcc as compared to infected patient serum. In fact, Lindenbach and colleagues, PNAS 2006, reported a 100x lower specific infectivity for cell culture virus as compared to virus retrieved from animals. The relative proportions of subgenomic, naked, encapsidated, enveloped, lipoviroparticle-associated, exosome-associated viral RNA might be completely different from what is found in patient serum samples. On the other hand, the cancer cell lines in culture might behave differently than hepatocytes in clearing HCV genome, whether by uptake or RNase degradation inside or outside the cells. Would any other cancer cell line give the same clearance effect? If yes, how does it support the authors’ point: does the liver clear many viruses simply because it is very vascularized?

The in vitro experiment is provided to support what is primarily an in vivo study. The Reviewer acknowledges “the low impact of the cell culture experiment on the authors conclusions” (see Reviewer comment #2 below), which is simply that hepatocytes may be involved in viral clearance, and that additional investigation is needed (lines 13-14 and 113, respectively). We believe this relegates the limitations of the in vitro experiment to a relatively minor issue. We provide the in vitro data because it surprisingly parallels what was observed in vivo and feel this is worth conveying.

There is no doubt that there are differences between in vitro and in vivo systems and we have done our best to clearly highlight this fact and note the extensive further analysis required before the cell culture data would allow final conclusions (lines 193-195; 200-209). However, to address some of the specific Reviewer comments, it is well known in the field that it is primarily the lipoprotein/lipid differences between the in vitro and in vivo HCV particles that cause the different specific infectivity (and not issues of subgenomic RNA or naked RNA). There are certainly different levels of exosomes in vitro and in vivo (and between cell types) but exosomes are a minor species in both cases. As such, we believe including this preliminary, but consistent, data from the widely used cell culture system strengths the in vivo data and is exciting because it suggests that we may indeed have a means to dissect this further in vitro, and importantly it opens up the possibility for the entire HCV research community to contribute to this effort. However, dissecting this further is a major undertaking which is beyond the scope of this paper. It is in fact the focus of an entire Aim in our recent NIH R01 renewal and will take years to complete (e.g. different cell lines, different primary cells, organoids, other viruses, different sources of HCV, etc.).

Because this is not a simple issue that can be addressed fully at this time, we have instead revised the manuscript to further explain the preliminary and purely supportive role of the in vitro experiment and discuss in more detail the obvious limitations and further work that might allow for the dissecting of this phenomena in vitro (lines 193-195; 200-209).

2. I agree that the definition of biological replicates is problematic and subject to interpretation in particular with cell lines. However, as stated in the authors’ response, biological replicates refer to “biologically distinct samples“. If I understood well, the “biological duplicates“ described in Figure 4 are replicate wells of the Huh7 cell line that were seeded, treated and infected in parallel, with the same cell passage, virus and inhibitor stocks, on the same day, which to me does not quality as biologically distinct samples, and the variation observed between these wells is mostly technical (as opposed to different mice in an animal experiment for instance). I therefore recommend removing “biological duplicate“ from the figure legend. Since the data shown is representative of 2 experiments, the conclusion would be stronger by averaging the 2 independent experiments.Given the low impact of the cell culture experiment on the authors conclusions and its strong limitations, I recommend the authors significantly reinforce the in vitro evidence and discuss the remaining limitations more in detail (as discussed above and in the previous review: testing other cell types, checking infectivity in addition to genome copies, etc).

Respectfully, while the reviewer may not agree with the definition of biological replicate that is standard in the field (and defined by the NIH), that is the definition we are using. According to the common/NIH definition, technical replicates are when the same sample is measured multiple times (e.g., our qPCR replicates, which represent random noise associated with protocols or equipment). Biological replicates are parallel measurements of biologically distinct samples (e.g., our replicate wells, which capture random biological variation in replicate samples that are then measured in parallel). That being said, we have no issues removing the phrase “biological replicates” from the figure legend.

Additionally, as requested, we now showing the average of 2 no cell experiments in Figure 4A and the average of 3 daclatasvir experiments and 2 naringenin experiments in Figure 4B. We had already averaged the one representative DCV and NG experiments, but now average all five of the DCV and NG experiments. The data from all the individual experiment is provided in Figure 4 source data file. This should make it clear that the results are reproducible in multiple experiments even when two different inhibitors are used to block HCV secretion. We believe the additional data provided should put to rest any concerns about the data reproducibility/variability.

Reviewer #3:Figure 1 could be made of 2 figures: 1/ the existing one 2/ a zoom in in the first few hours.

We have added the figure (i.e., Fig. 1B in the revised manuscript) as suggested by the referee and referenced the revised subpanels within the text (lines 39-40) and Figure 1 legend.